# Complementing a Policy with a Different Observation Space

Gokul Swamy [1]   Sanjiban Choudhury [2]   J. Andrew Bagnell [3 1]   Zhiwei Steven Wu [1]

## Abstract

We consider the problem of improving upon a black-box policy which operates on a different observation space than the learner. Such problems occur when augmenting an existing hand-engineered system with a new machine learning model or in a shared autonomy / human-AI complementarity context. We prove that following the naive policy gradient can lead to a decrease in performance because of incorrect grounding in a different observation space. Then, if we have access to both sets of observation at train time, we derive a method for correctly estimating a policy gradient via an application of backdoor adjustment. If we don't, we prove that under certain assumptions, we can use the proxy correction to correctly estimate a direction of improvement.

## 1. Introduction

Classical results tell us that in a Partially Observed Markov Decision Process (POMDP), finding high-value policies that map observations to actions is computationally hard (Papadimitriou & Tsitsiklis, 1987). Furthermore, a naive application of MDP techniques (e.g. value-iteration, policy-iteration, or $Q$-learning) empirically leads to unstable and often decaying performance over iterations (Singh et al., 1994; Littman, 1994). However, folk wisdom suggests that if one computes (or approximates well) $Q^\pi(o, a)$ – the long-term expected return of an observation-action pair – and then makes "small adjustments" to the policy to select higher value actions for a given observation, the policy can be reliably improved. Such small changes can then be repeated to achieve a notion of local optima. This folk wisdom finds precision for stochastic mixtures of policies (Jaakkola et al., 1994; Kakade & Langford, 2002; Daumé et al., 2009), parameterized stochastic policies (Baxter & Bartlett, 2001;

---

*Equal contribution [1]Carnegie Mellon University [2]Cornell University [3]Aurora Innovation. Correspondence to: Gokul Swamy <gswamy@cmu.edu>.

*Proceedings of the Interactive Learning with Implicit Human Feedback Workshop at ICML 2023.*, Honolulu, Hawaii, USA. PMLR 202, 2023. Copyright 2023 by the author(s).

Williams, 1992), and when searching over a set of deterministic policies (Bagnell et al., 2003).

An oft-underemphasized requirement for the above policy improvement theorems to hold is that the proposed change to the policy uses *the same observation space* as the policy it is improving over. In practice, this often doesn't hold, as ML practitioners may be attempting to improve a policy that comes from an existing, effectively black-box system to them, or even to provide action recommendations to expert humans who operate with a fundamentally different observation space.

For example, consider a medical assistance system to help doctors at a walk-in clinic. In response to observing inflamed tonsils, a doctor might prescribe an antibiotic. The automated system, which operates purely based on numerical information, does not have access to this feature but does observe that patients always get better after being prescribed amoxicillin. It would then over-estimate $Q^{\text{doctor}}(o, \text{amoxicillin})$, leading to it suggesting uneccesary prescriptions of a strong medication that could have unfortunate side-effects for patients with issues other than strep throat. More generally, naively computing $Q^\pi(o, a)$ where $\pi$ uses a different observation space than $o$ can lead to estimates of cumulative reward that will not match what the learner would actually receive at test time.

In this work, we study the perhaps initially surprising failure of these policy improvement algorithms. We identify their fundamental issue: ***when observations spaces differ, $Q^\pi(o, a)$ becomes causally confounded, rendering the standard Monte-Carlo estimate inconsistent.*** We then study two regimes- - first where a richer observation space is available when we train a policy, but not when we execute that policy ; second, a more difficult setting where at both train and test we have access only to a history of an observation space more limited than the initial policy we wish to improve over. Our technical tools leverage recent advances in proxy variable correction techniques (Miao et al., 2018; Tchetgen et al., 2020) and approximate back-door corrections (Pearl, 2009; Xu & Gretton, 2022) to provide correct policy gradients in confounded settings.

More explicitly, our work makes the following three contributions.

**1. We formalize the problem of improving upon a policy with a different observation space.** We prove that correct estimation of $Q^\pi(o, a)$ values is sufficient for policy improvement. We then give a family of examples under which the standard, Monte-Carlo estimate for $Q^\pi(o, a)$ will not lead to policy improvement.

**2. We consider the setting when both sets of observations are available at training time.** We show how we can use Pearl's backdoor adjustment (Pearl, 2009) to de-confound our $Q$-value estimates. We derive a procedure efficiently approximating the backdoor adjustment using the output of a simple classifier.

**3. We consider the setting in which the learner never observes the features the baseline policy used to make decisions.** Under some assumptions, we show how we can use the proxy correction (Miao et al., 2018; Tchetgen et al., 2020; Bennett & Kallus, 2021) to correct estimate $Q$-values in the learner's observation space. We derive two algorithms for doing so, one of a generative-modeling flavor and one of a game-theoretic flavor.

We now turn our attention to formalizing the problem we consider.

## 2. Formalism

Consider a POMDP (Kaelbling et al., 1998) with two different observation spaces, $\mathcal{O}_1$ and $\mathcal{O}_2$. We can represent our setup via the following structural causal model (SCM).

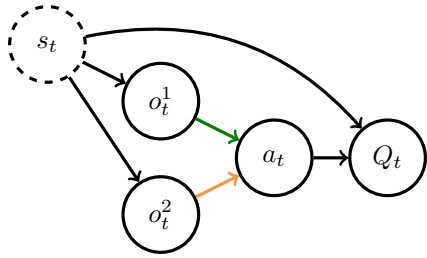

*Figure 1.* The causal model we consider in this paper. The green arrow denotes the dependence in the training data, the orange arrow denotes the dependence of the policy we're learning.

Let

$$J(\pi_\theta) = \mathbb{E}_{\xi \sim \pi_\theta} \left[ \sum_t^T r(s_t, a_t) \right] \quad (1)$$

denote the performance of a policy. We assume we see data generated by some behavior policy $\pi_b : \mathcal{O}^1 \to \Delta(\mathcal{A})$ that consists of tuples of $(o_t^2, a_t, Q_t)$ and would like to find a direction in policy-space that improves $J(\pi_b)$. More formally, we're looking for a policy $\pi : \mathcal{O}^2 \to \Delta(\mathcal{A})$ such that

$$J((1 - \epsilon)\pi_b + (\epsilon)\pi) \geq J(\pi_b) \quad (2)$$

for small enough $\epsilon$.

### 2.1. Policy Improvement in POMDPs

The first complexity we have to deal with is that our policies depend on observations rather than states. Following an argument similar to that of Jaakkola et al. (1994), we prove the following theorem.

**Theorem 2.1.** *Define* $\pi_\epsilon = (1 - \epsilon)\pi_b + (\epsilon)\pi$, *where* $\pi_b$ *operates on* $\mathcal{O}^2$ *and* $\pi$ *operates on* $\mathcal{O}^1$. *Then, we have that*

$$J(\pi_\epsilon) - J(\pi_b) =$$
$$\epsilon T \mathbb{E}_{o \sim \rho_{\pi_\epsilon}} [\mathbb{E}_{a \sim \pi(a|o)} [\mathbb{E}_{s \sim \rho_{\pi_b}(s|o)} [A^{\pi_b}(s, a)]]] + O(\epsilon^2 T^2).$$
$$(3)$$

Define

$$A^{\pi_b}(o, a) = \mathbb{E}_{s \sim \rho_{\pi_b}(s|o)} [A^{\pi_b}(s, a) - V^{\pi_b}(s)], \quad (4)$$

and $Q^{\pi_b}(o, a)$, $V^{\pi_b}(o)$ analogously. The above result tells us that as long as we are able to estimate $Q^{\pi_b}(o, a)$ values correctly, putting more probability mass on actions with positive advantages with a step of size $\epsilon < O(\frac{1}{T})$ is sufficient to guarantee improvement.

For comparison, consider the (vanilla) policy gradient for MDPs.

$$\nabla_\theta J(\pi_\theta) = \mathbb{E}_{\xi \sim \pi_\theta} \left[ \sum_t^T \nabla_\theta \log(\pi_\theta(a|s)) Q^{\pi_\theta}(s, a) \right]. \quad (5)$$

One usually subtracts a baseline to reduce variance. [1] The optimal control variate is the value function (Sutton & Barto, 2018), which gives us the following expression

$$\nabla_\theta J(\pi_\theta) = \mathbb{E}_{\xi \sim \pi_\theta} \left[ \sum_t^T \nabla_\theta \log(\pi_\theta(a|s)) A^{\pi_\theta}(s, a) \right]. \quad (6)$$

Similar to our above result, the theory of policy gradients (Kakade & Langford, 2002; Agarwal et al., 2020) tells us that mixing in a small enough portion of the greedily optimal policy is sufficient to guarantee improvement. Thus, the main difference comes in the definition of the advantage function: for MDPs, advantages depend on states while for our setup, advantages depend on the observation space of the policy we're mixing in. We therefore turn our attention to estimating $Q$-values / advantages.

---

[1]One usually pre-conditions with the Fisher information matrix to follow the natural policy gradient (Kakade, 2001), but we ignore those details for simplicity.

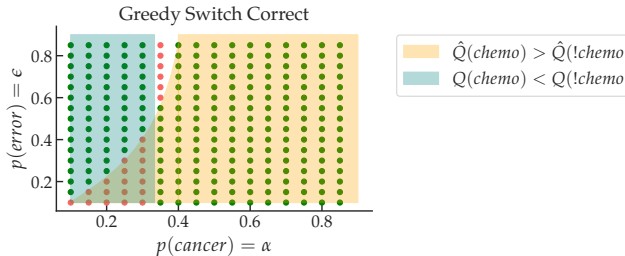

*Figure 2.* We see that when the base rate of cancer is low and the doctor makes few mistakes, we under-estimate the value of chemo and would end up providing poor care to people. This is indicated in the figure by the red dots.

## 2.2. An Example of why the Naive $Q$-value Estimate Fails

We illustrate the issue with the naive $Q$-value estimator with a simple, one-step example.

**Problem 2.2** (Cancer Treatment). Consider the following one-step (contextual bandit) problem. A doctor observes whether a patient has cancer ($\mathcal{O}^1 = \{$cancer, !cancer$\}$) and decides whether to instruct them to begin chemotherapy. The rewards are as follows:

$$Q(\text{cancer}, \text{chemo}) = 1, Q(\text{!cancer}, \text{chemo}) = 0, \quad (7)$$
$$Q(\text{cancer}, \text{!chemo}) = 0, Q(\text{!cancer}, \text{!chemo}) = 0.5$$

We assume $\alpha$ fraction of the population has cancer. The doctor makes a mistake (i.e. picking the action with the lower $Q$ value) with probability $\epsilon$. We want to learn a policy that, without any observations, improves upon the doctor ($\mathcal{O}^2 = \emptyset$).

To solve this problem, we compute both $Q(\emptyset, \text{chemo})$ and $Q(\emptyset, \text{!chemo})$ by averaging the rewards received when a patient was / wasn't given chemotherapy, regardless of their underlying cancer status. More formally,

$$\hat{Q}(\emptyset, a) = \mathbb{E}_{\pi_b}[Q(S, a)] \quad (8)$$

We pick the action with the higher $Q$ value always (i.e. $\pi(\emptyset) = \max_a Q(\emptyset, a)$). We visualize the region of $(\alpha, \epsilon)$ in which we pick !chemo in yellow in the below figure. Given access to the base rate of cancer ($\alpha$), we can also compute the ground-truth $Q$ values, which tell us when giving no patients chemotherapy would be preferable to giving all of them chemotherapy. Intuitively, this is true when few patients have cancer. We visualize this in teal in the figure.

Mixing in *any* amount of the greedy optimal policy is harmful when $\hat{Q}$ and $Q$ rank the two actions in the opposite order, which correspond to the red dots in the above figure. Intuitively, the cluster of red dots in the bottom left of the

figure corresponds to the learner under-estimating the value of chemotherapy when few people need it and doctors rarely spuriously prescribe it.

If such a problem can happen even in a single-step problem, it can clearly happen with multiple steps of interaction. We therefore turn our attention to using techniques from causal inference to de-bias our $Q$-value estimates.

## 2.3. Estimating $Q$-values when both Observations are Available

We first consider the relatively simple setting when both sets of observations are provided to the learner in the behavioral data. Formally, we want to compute

$$Q(o_t^2, a) = \mathbb{E}[\sum_{\tau=t}^{T} r(s_\tau, a_\tau)|do(a_t), o_t^2]. \quad (9)$$

Using $Q_t$ as a shorthand for the sum of rewards over the horizon, we can apply the rules of do-calculus (equivalently, the backdoor adjustment formula (Pearl, 2009)) to compute the above causal effect.

$$\mathbb{E}[Q_t|do(a_t), o_t^2] = \sum_{o_t^1} P(o_t^1|do(a_t), o_t^2)\mathbb{E}[Q_t|do(a_t), o_t^1, o_t^2]$$
$$= \sum_{o_t^1} P(o_t^1|do(a_t), o_t^2)\mathbb{E}[Q_t|a_t, o_t^1, o_t^2]$$
$$= \sum_{o_t^1} P(o_t^1|o_t^2)\mathbb{E}[Q_t|a_t, o_t^1, o_t^2],$$

where the second equality follows from $(Q_t \perp\!\!\!\perp a|o_t^1, o_t^2)_{\mathcal{G}_{\underline{a_t}}}$ and Rule 2 and the third equality follows from $(o_t^1 \perp\!\!\!\perp a|o_t^2)_{\mathcal{G}_{\overline{a_t}}}$ and Rule 3. Lastly, simple algebra tells us that

$$\mathbb{E}[Q_t|do(a_t), o_t^2] = \mathbb{E}_{o_t^1}\left[\frac{P(o_t^1, o_t^2)}{P(o_t^1)P(o_t^2)}\mathbb{E}[Q_t|a_t, o_t^1, o_t^2]\right]$$
$$= \mathbb{E}_{o_t^1}\left[r(o_t^1, o_t^2)\mathbb{E}[Q_t|a_t, o_t^1, o_t^2]\right], \quad (10)$$

where we use $r(o_t^1, o_t^2)$ to denote the density ratio between the joint and the product of marginals.

The above expression gives us the following procedure for policy improvement.

1. Estimate $\mathbb{E}[Q_t|a_t, o_t^1, o_t^2]$ via regression.

2. Fit a density ratio estimate for $r(o_t^1, o_t^2)$.

3. Fit $Q(o_t^2, a_t)$ via a secondary regression with targets determined via the previous two components.

4. Compute the greedily optimal policy and plug into your favorite policy improvement method to control the step size.

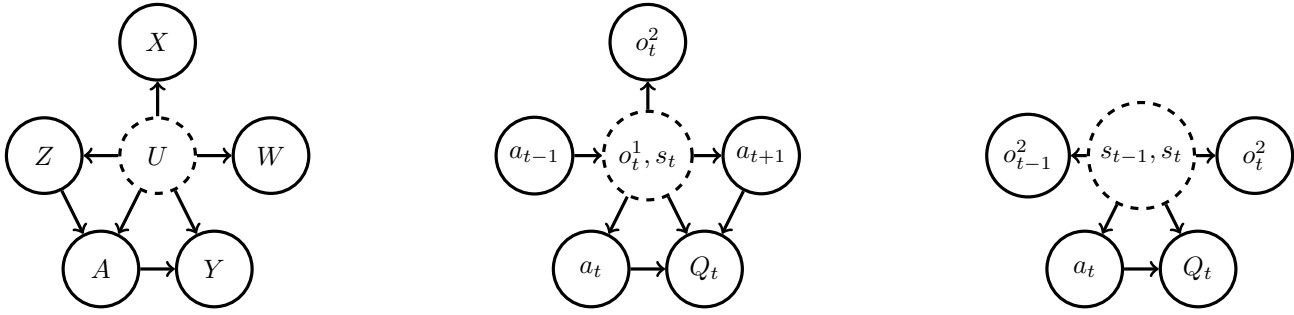

*Figure 3.* Causal models that satisfy the independence assumptions required to apply the proxy correction.

We can perform the seconds step via classification between paired and unpaired samples of $o_t^1, o_t^2$ (Hyvarinen & Morioka, 2016).

## 2.4. Estimating $Q$-values with only Learner Observations

We first describe the general technique we use before specializing to our case of POMDPs. The argument we present below is a slight generalization of that in Xu et al. (2021).

### 2.4.1. THE PROXY CORRECTION

We are attempting to determine causal effect $\mathbb{E}[Y|do(a), x]$. Unfortunately, $Y, A, X$ all have a common parent: unobserved confounder $U$. However, we have access to two "proxies", $W, Z$, for the unobserved $U$ that we can use to estimate the effect. Formally, we require the following independence conditions (Miao et al., 2018):

$$\text{C1: } W \perp\!\!\!\perp (A, Z)|U, X \tag{11}$$

$$\text{C2: } Y \perp\!\!\!\perp Z|A, U, X \tag{12}$$

An example of a graph that satisfies these conditions is drawn in Figure 3, left. Consider the following set of conditional moment restrictions (CMR) on $g$:

$$\mathbb{E}[Y|a, z, x] = \mathbb{E}[g(a, x, W)|a, z, x], \forall (a, z, x) \in (\mathcal{A} \times \mathcal{Z} \times \mathcal{X}). \tag{13}$$

We prove the following theorem.

**Theorem 2.3.** *Define*

$$h(a, x) = \mathbb{E}_{W|x}[g(a, x, W)]. \tag{14}$$

*If $g$ satisfies Equation 13, then $\mathbb{E}[Y|do(a), x] = h(a, x)$.*

Swamy et al. (2022) provide two methods for learning a $g$ that satisfies the CMR in one of two ways. The first is via game-solving:

$$\min_g \max_f \mathbb{E}[f(A, X, Z)(Y - g(A, X, W)) - 0.5f^2(A, X, Z)].$$

The second is via generative modeling. First, we learn a model of $P(W|a, z, x)$. Then, we minimize the following MSE

$$\min_g \mathbb{E}_{A, Z, X}[(\mathbb{E}[Y|a, z, x] - \mathbb{E}[g(a, x, W)|a, z, x])^2]. \tag{15}$$

Importantly, by drawing separate samples of $W$ for both evaluations of $g$ inside the square, we avoid the so-called "double-sample" problem that can lead to inconsistent estimates of a causal effect (Baird, 1995).

A third family of techniques, described by Xu et al. (2021), alternates between solving two regression problems, each of which corresponds to one of the stages of the above generative modeling approach. More formally, we assume that we can write $g$ and the conditional feature mean as

$$g(a, x, w) = u^T(\Psi_{A(2)}(a) \otimes \Psi_{X(2)}(x) \otimes \Psi_W(w))$$

$$\mathbb{E}_{W|a, z, x}[\Psi_W(w)] = V(\Psi_{A(1)}(a) \otimes \Psi_Z(z) \otimes \Psi_{X(1)}(x)),$$

where all $\Psi$ are learned neural networks and $u$ and $V$ are a vector and matrix, respectively. Observe that because we take a tensor product $\otimes$ between the output of neural network features which can be arbitrarily high dimensional, the above separation into functions of each random variable is without meaningful restriction on functions representable. We then alternate between minimizing two loss functions over subsets of the parameters. We refer interested readers to Xu et al. (2021)'s excellent paper for more details.

### 2.4.2. APPLYING THE PROXY CORRECTION TO POMDPs

Due to space concerns, we only write out the full form of the game-solving variant.

**Approach 1.** Consider the SCM in Figure 3, center.

Observe that we satisfy the requisite independence assumptions:

$$\text{C1: } a_{t-1} \perp\!\!\!\perp (a_t, a_{t+1})|s_t, o_t^2, o_t^1 \tag{16}$$

$$\text{C2: } Q_t \perp\!\!\!\perp a_{t-1}|a_t, s_t, o_t^2, o_t^1 \tag{17}$$

Putting it all together, we suggest the following method for policy improvement

1. Solve game

$$\min_g \max_f \mathbb{E}[f(a_t, o_t^2, a_{t-1})(Q_t - g(a_t, o_t^2, a_{t+1})) - 0.5f^2(a_t, o_t^2, a_{t-1})]$$

2. Fit generative model of $P(a_{t+1}|o_t)$.

3. Fit a model of $Q^{\pi_b}(o_t^2, a_t) = \mathbb{E}_{a_{t+1}|o_t}[g(a_t, o_t, a_{t+1})]$ via regression.

4. Compute the greedily optimal policy and plug into a policy improvement method to control the step size.

Note that the generative model we have to fit in the second step is just a policy.

**Approach 2.** An alternative approach which would allow us to estimate the same quantity would be to use the past and current observations as proxies (Bennett & Kallus, 2021). We visualize this approach graphically in Figure 3, right.

Note that because we directly use the observation as one of the proxies, we don't need to perform a secondary averaging. We can solve the following game to recover $Q_t$:

$$\min_g \max_f \mathbb{E}[f(a_t, o_{t-1})(Q_t - g(a_t, o_t)) - 0.5f^2(a_t, o_{t-1})] \tag{18}$$

After solving for $g$, we can directly plug it into an advantage estimation procedure (Schulman et al., 2018), allowing us to compute causally correct policy gradients.

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

# A. Proofs

## A.1. Proof of Theorem 2.1

*Proof.* Via the performance difference lemma (Kakade & Langford, 2002),

$$J(\pi_\epsilon) - J(\pi_b) = T\mathbb{E}_{a,o\sim\rho_{\pi_\epsilon}}[\mathbb{E}_{s\sim\rho_\epsilon(s|o)}[A^{\pi_b}(s,a)]]. \tag{19}$$

Fix $o \in \mathcal{O}^2$. Consider

$$
\begin{aligned}
\mathbb{E}_{a\sim\rho_{\pi_\epsilon}(o)}[\mathbb{E}_{s\sim\rho_\epsilon(s|o)}[A^{\pi_b}(s,a)]] &= \sum_{a\in\mathcal{A}}(\pi_b(a|o) + \Delta(a|o))\sum_{s\in\mathcal{S}}(\rho_{\pi_b}(s|o) + \Delta(s|o))(Q^{\pi_b}(s,a) - V^{\pi_b}(s)) \\
&= \sum_{a\in\mathcal{A}}\Delta(a|o)\sum_{s\in\mathcal{S}}(\rho_{\pi_b}(s|o) + \Delta(s|o))(Q^{\pi_b}(s,a) - V^{\pi_b}(s)) \\
&= \epsilon\sum_{a\in\mathcal{A}}\pi(a|o)\sum_{s\in\mathcal{S}}(\rho_{\pi_b}(s|o) + \Delta(s|o))(Q^{\pi_b}(s,a) - V^{\pi_b}(s)) \\
&= \epsilon\sum_{a\in\mathcal{A}}\pi(a|o)\sum_{s\in\mathcal{S}}(\rho_{\pi_b}(s|o))(Q^{\pi_b}(s,a) - V^{\pi_b}(s)) + O(\epsilon^2 T^2).
\end{aligned}
$$

The last line comes from the fact that

$$d_{TV}(\rho_{\pi_b}(s|o), \rho_{\pi_\epsilon}(s|o)) \leq \epsilon T, \tag{20}$$

as proved in Agarwal et al. (2019).

Taking the average over observations tells us that

$$J(\pi_\epsilon) - J(\pi_b) = \epsilon T\mathbb{E}_{o\sim\rho_{\pi_\epsilon}}[\mathbb{E}_{a\sim\pi(a|o)}[\mathbb{E}_{s\sim\rho_{\pi_b}(s|o)}[A^{\pi_b}(s,a)]]] + O(\epsilon^2 T^2). \tag{21}$$

$\square$

## A.2. Proof of Theorem 2.3

*Proof.* Via backdoor adjustment (Pearl, 2009),

$$\mathbb{E}[g(a,x,W)|a,z,x] = \mathbb{E}_{U|a,z,x}[\mathbb{E}[g(a,x,W)|a,z,x,u]] = \mathbb{E}_{U|a,z,x}[\mathbb{E}[g(a,x,W)|x,u]], \tag{22}$$

where the second equality comes from assumption C1. Next, we note that

$$\mathbb{E}[Y|a,z,x] = \mathbb{E}_{U|a,z,x}[\mathbb{E}[Y|a,z,x,u]] = \mathbb{E}_{U|a,z,x}[\mathbb{E}[Y|a,x,u]], \tag{23}$$

where the second equality comes from assumption C2. Now, if we satisfy the CMR uniformly, we know that

$$\mathbb{E}_{U|a,z,x}[\mathbb{E}[g(a,x,W)|x,u] - \mathbb{E}[Y|a,x,u]] = 0. \tag{24}$$

If we assume "completeness" of our confounder (Miao et al., 2018; Tchetgen et al., 2020), this implies that the above condition implies the following pointwise guarantee:

$$\mathbb{E}[g(a,x,W)|x,u] = \mathbb{E}[Y|a,x,u]. \tag{25}$$

This guarantee allows us to plug the LHS into the backdoor adjustment formula for the causal effect. More formally,

$$\mathbb{E}[Y|do(a),x] = \mathbb{E}_{U|x}[\mathbb{E}[Y|a,u,x]] = \mathbb{E}_{U|x}[\mathbb{E}[g(a,x,W)|x,u]] = \mathbb{E}_{W|x}[\mathbb{E}[g(a,x,W)]], \tag{26}$$

where the first equality comes from the backdoor adjustment formula and the last from the law of iterated expectation. $\square$

