# OpenReview forum: "Complementing a Policy with a Different Observation Space"
_ICML.cc/2023/Workshop/ILHF — ILHF Workshop ICML 2023_

### Official Review · Reviewer_XZ8s · 2023-06-06
**Good motivation and structure, grounding the proofs in examples would be helpful**

**Rating:** 6
**Confidence:** 2

**Review:**

This paper addresses the problem of learning a policy under a specific observation space by improving upon a different policy operating on a different observation space. The authors begin by motivating the problem, using a toy example derived from cancer treatment. Then, they propose two methods of learning the policy on the new observation space, each under one of the following cases: 1) both observation spaces are available during learning, 2) only the new observation space is available during learning.

Overall the paper is well-motivated and addresses a relevant problem. The structure of the paper is very clear, but at some points the equations are hard to follow. Some specific suggestions of improvement for the paper:

- It would be very helpful if the authors could provide toy examples following their recommended procedures in section 2.3 and 2.4.2 that help ground their theoretical analysis.
- In the procedure of section 2.3, can you expand on "targets determined via the previous two components"?
- In section 2.3, can you explain why $Q_t \perp a_t | o_t^1, o_t^2$?
- Why there is a causal relationship from $a_{t+1}$ to $Q_t$ in Figure 3? It makes sense that $Q_t$ depends on $a_t$, but I don't understand why it depends on $a_{t+1}$
- A conclusion and outlook to future work are missing.

Other comments:

- Define $\Delta$ and $\xi$ in equation 2
- Theorem 2.1 presents the operation spaces in an inverse way with respect to the paragraph right before equation 2
- Theorem 2.1 Lacks a reference to the appendix
- Equation 4 --> The inner $A^{\pi_b}(s,a)$ should be $Q^{\pi_b}(s,a)$?
- Figure 2 --> define red dots and green dots
- In paragraph before equation 8 --> $Q$ should be $\hat{Q}$?
- "we pick !chemo in yellow in the below figure" in paragraph after equation 8 --> Shouldn't it be chemo (without !)?
- Define $do(a_t)$ in equation 9
- Define $\mathcal{G}_{a_t}$ (note the bar is sometimes on top of $a_t$, other times under it)
- Spell out SCM

---

### Official Review · Reviewer_85MB · 2023-06-15
**Complementing a Policy with a Different Observation Space**

**Rating:** 5
**Confidence:** 2

**Review:**

Originality:
The contributions in this paper seem original, however, I don't see a related work section that positions this paper with the literature, and hence hard to gauge the significance of this work.

Summary:
This paper considers the problem of improving upon a black-box policy operating on a different observation space than the learner.  In this case, naive policy gradient fails due to observation mismatch. This paper proposes using a backdoor adjustment to estimate the policy gradient when both sets of observations are available during train time. In the case the observation pairs are not available during training, under certain assumptions, a direction of improvement can be correctly estimated. Following this, the paper makes 3 contributions. 1) formalizing the problem statement of policy improvement under mismatched observation space, 2) How to improve policy when both sets of observations are available during training, and 3) What if the learner cannot observe the features of the behavior policy.

Comments: The paper's contributions lean too much on the theoretical side that it has failed to show reasonable experimental results, I think the paper could have focused a bit more on experiments to showcase the theoretical contributions, which seems lacking in my opinion. In section 2.4, it's hard to understand how these contributions translate to estimating Q with only learner observations. The paper also ends abruptly without a limitations and conclusions section.

---

### Decision · Program_Chairs · 2023-06-20

Accept